# New Metabolites from the Marine Sponge *Scopalina hapalia* Collected in Mayotte Lagoon

**DOI:** 10.3390/md20030186

**Published:** 2022-03-02

**Authors:** Charifat Saïd Hassane, Gaëtan Herbette, Elnur Garayev, Fathi Mabrouki, Patricia Clerc, Nicole J. de Voogd, Stephane Greff, Ioannis P. Trougakos, Jamal Ouazzani, Mireille Fouillaud, Laurent Dufossé, Béatrice Baghdikian, Evelyne Ollivier, Anne Gauvin-Bialecki

**Affiliations:** 1Laboratoire de Chimie et de Biotechnologie des Produits Naturels, Faculté des Sciences et Technologies, Université de La Réunion, 15 Avenue René Cassin, CS 92003, CEDEX 09, 97744 Saint-Denis, France; charifat.saidhassane@gmail.com (C.S.H.); patricia.clerc@univ-reunion.fr (P.C.); mireille.fouillaud@univ-reunion.fr (M.F.); laurent.dufosse@univ-reunion.fr (L.D.); 2CNRS, Centrale Marseille, FSCM, Spectropole, Aix Marseille Université, Campus de St Jérôme-Service 511, 13397 Marseille, France; gaetan.herbette@univ-amu.fr; 3CNRS 7263, IRD 237, IMBE, Aix Marseille Université, Avignon Université, 13331 Marseille, France; elnur.garayev@univ-amu.fr (E.G.); fathi.mabrouki@univ-amu.fr (F.M.); stephane.greff@imbe.fr (S.G.); beatrice.baghdikian@univ-amu.fr (B.B.); evelyne.ollivier@univ-amu.fr (E.O.); 4Naturalis Biodiversity Center, Darwinweg 2, 2333 CR Leiden, The Netherlands; nicole.devoogd@naturalis.nl; 5 Institute of Environmental Sciences, Leiden University, Einsteinweg 2, 2333 CC Leiden, The Netherlands; 6Department of Cell Biology and Biophysics, Faculty of Biology, National and Kapodistrian University of Athens, 15784 Athens, Greece; itrougakos@biol.uoa.gr; 7Institut de Chimie des Substances Naturelles, CNRS UPR 2301, Université Paris-Saclay, 1, Av. de la Terrasse, 91198 Gif-sur-Yvette, France; jamal.ouazzani@cnrs.fr

**Keywords:** *Scopalina hapalia*, butenolides, lipids, diketopiperazines, molecular networking, bromotyrosine

## Abstract

The biological screening of 44 marine sponge extracts for the research of bioactive molecules, with potential application in the treatment of age-related diseases (cancer and Alzheimer’s disease) and skin aging, resulted in the selection of *Scopalina hapalia* extract for chemical study. As no reports of secondary metabolites of *S. hapalia* were found in the literature, we undertook this research to further extend current knowledge of *Scopalina* chemistry. The investigation of this species led to the discovery of four new compounds: two butenolides sinularone J (**1**) and sinularone K (**2**), one phospholipid 1-*O*-octadecyl-2-pentanoyl-*sn*-glycero-3-phosphocholine (**3**) and one lysophospholipid 1-*O*-(3-methoxy-tetradecanoyl)-*sn*-glycero-3-phosphocholine (**4**) alongside with known lysophospholipids (**5** and **6**), alkylglycerols (**7**–**10**), epidioxysterols (**11** and **12**) and diketopiperazines (**13** and **14**). The structure elucidation of the new metabolites (**1**–**4**) was determined by detailed spectroscopic analysis, including 1D and 2D NMR as well as mass spectrometry. Molecular networking was also explored to complement classical investigation and unravel the chemical classes within this species. GNPS analysis provided further information on potential metabolites with additional bioactive natural compounds predicted.

## 1. Introduction

Marine sponges are known as an extensive source of novel natural products with a wide variety of biological activities [1]. As a hotspot of biodiversity, the Comoros Archipelago, situated off the southeastern coast of Africa, in the Mozambique channel, is poorly explored for natural products chemistry [2]. Hence, the archipelago represents a great potential for the discovery of original biologically active metabolites [3,4]. In the course of our research aimed at the discovery of bioactive metabolites with potential application in the treatment of age-related diseases and skin aging such as anti-CDK7, anti-tyrosinase or sirtuin 1 activators substances, 44 marine sponges were collected from the lagoon of Mayotte and evaluated. This screening effort revealed 19 marine sponges potentially producing anti-aging metabolites. These bioactive marine sponges include highly studied genera such as *Haliclona* [5] and *Jaspis*, as well as less studied ones like *Scopalina* and *Svenzea*. Amongst this last category, the *Scopalina hapalia* (Order Scopalinida) [6] extract exhibited significant inhibitory activities against the protein kinase FynB and elastase, respectively, involved in neurodegenerative diseases [7,8] and skin aging [9]. Therefore, it was selected for further chemical investigation. This extract gained special attention mainly as, at this moment, only six metabolites were reported from a specimen of *S. ruetzleri* (formerly known as *Ulosa ruetzleri*) collected in Harrington Sound, Bermuda [10]. The first compounds reported were an unusual lipid 17*Z*-tetracosenyl 1-glycerol ether along with batyl alcohol and 4-(4-hydroxyphenyl)-3-buten-2-one [10]. However, the last compound is suspected to be an artifact derived from the aldol condensation of the extraction solvent acetone with *p*-hydroxybenzaldehyde [10]. In 1986, the new compound 4-hydroxy-5-(indole-3-yl)-5-oxo-pentan-2-one with plant regulatory activity was isolated [11]. The last two novel compounds reported were a phosphorylated hydantoin, ulosantoin, and a structurally related compound, dimethyl N2-creatininylphosphate [12]. While ulosantoin exhibited significant insecticidal property, dimethyl N2-creatininylphosphate showed no activity in the insecticidal screening [12]. More recently, organic extracts or fractions of *S. ruetzleri* collected near Brazilian and Colombian coasts exhibited anti-proliferative effects on human breast cancer cells [13], antifungal effects against *Candida albicans* [14], antioxidant effect [15] as well as cytotoxic activities against human glioma and neuroblastoma cell lines [15]. An ethanolic extract of *S. hapalia* collected from the reefs of Zanzibar also demonstrated antibacterial and concentration-dependent cytotoxic activities [16] opening up interesting perspectives for this understudied species. The species *Scopalina hapalia* is largely distributed in the southwestern region of the Indian Ocean (Zanzibar, Mayotte). It was also reported in Australia, Papua New Guinea [17] and Singapore [18]. 

In this study, we report the isolation of two new 2,3-dimethyl butenolide derivatives (**1** and **2**), one new phospholipid (**3**), one new lysophospholipid (**4**) alongside with known lysophospholipids (**5** and **6**), alkylglycerols (**7**–**10**), epidioxysterols (**11** and **12**) and diketopiperazines (**13** and **14**) (Figure 1). While the extract showed inhibition activity in the Fyn kinase and elastase bioassays, indicating the presence of bioactive compounds, most of the isolated compounds were obtained in insufficient amounts for biological testing. Contrary to our expectations, the few tested compounds (**6**, **8**, **9**) did not show significant biological activity. This paper reports the structural elucidation of the new compounds. Further, in an attempt to thoroughly assess the molecular potential of this less studied genus, we use a GNPS molecular networking approach. This approach revealed several chemical classes, among which some were isolated.

## 2. Results and Discussion

### 2.1. Chemical Investigation

Sponge specimens were extracted with CH_2_Cl_2_/CH_3_OH (1:1) and the resulting extracts were evaluated against molecular targets involved in age-related diseases and skin aging. The CH_2_CL_2_/CH_3_OH extract of *Scopalina hapalia* was found to inhibit 48% and 44% of the elastase activity, respectively, at the concentration of both 10 µg/mL and 1 µg/mL. The same extract also inhibited the activity of the protein kinase FynB in a concentration dependent manner at 111 µg/mL, 11.1 µg/mL, and 1.1 µg/mL. Consequently, the CH_2_CL_2_/CH_3_OH (1:1) extract was fractionated by MPLC, however, none of the obtained fractions exhibited anti-aging activities. Despite the loss of bioactivity, the chemical investigation was pursued. The CH_2_CL_2_/CH_3_OH extract was, first, subjected to partition between cyclohexane and CH_3_OH. The CH_3_OH-soluble portion was chromatographed over C-18 silica gel, followed by semipreparative C-18 silica HPLC purification, yielding compounds (**1**, **12**–**14**). In a second phase, the whole CH_2_CL_2_/CH_3_OH extract was also fractionated by MPLC over silica gel. Fractions of interest were further purified by either analytical or semipreparative C-18 silica HPLC to give metabolites (**2**–**11**).

The known compounds were identified by comparing their spectroscopic and/or spectrometric data with those reported in the literature (for structures and NMR data, see Appendix A). Compounds **5** and **6** were, respectively, identified as the known lysophospholipids 1-*O*-octadecyl-*sn*-glycero-3-phosphocholine [19] and 1-palmitoyl-*sn*-glycero-3-phosphocholine [20]. Compounds **7** to **10** correspond to known monoalkyl glycerol ethers previously isolated from marine sponges: 1-*O*-hexadecylglycerol (chimyl alcohol **7**) [21], 1-*O*-octadecylglycerol (batyl alcohol **8**) [21], 3-nonadecyloxy-1,2-propanediol (glycerol 1-nonadecyl ether **9**) [22] and 3-icosoxypropane-1,2-diol (glycerol 1-eicosyl ether **10**) [22]. Compounds **11** and **12** were determined, respectively, as a mixture of 5α,8α-epidioxy-24(*R*/*S*)-hydroperoxystigmasta-6,28-dien-3β-ol [23] and a mixture of 5α,8α-epidioxy-24(*R*/*S*)-stigmasta-6,22*E*-dien-3β-ol [24]. These two epidioxysterols were previously isolated from marine sponges *Lendenfeldia chondrodes* [23] and *Thalysias juniperina* [24]. The last two known compounds identified were diketopiperazines cyclo(Val-Leu) (**13**) and cyclo(Val-Phe) (**14**) [25]. These two compounds have previously been isolated from marine microorganisms *Streptomyces fungicidicus* and *Paecilomyces marquandii* [26,27].

Compound **1** was isolated as a yellow oil. High-resolution mass spectrometry (HRMS) analysis revealed a sodium adduct ion at *m*/*z* 349.1985 [M + Na]^+^ consistent with molecular formula C_18_H_30_O_5_ and incorporating four unsaturations. It provided a negative specific rotation ${\left[a\right]}_{589}^{25}-40$ (c 0.024, CH_3_OH). Analysis of the ^1^H NMR and HSQC data (Table 1) indicated that **1** contained two olefinic methyl singlets H-16 (δ_H_ 1.84) and H-17 (δ_H_ 1.90), one oxygenated methyl protons H-18 (δ_H_ 3.07), two methylene protons near heteroatom H-5 (δ_H_ 1.73, 1.95) and H-14 (δ_H_ 2.15) and eight methylene protons (δ_H_ 1.13−1.59). A comprehensive analysis of the ^13^C NMR revealed two olefinic carbons C-2 (δ_C_ 128.4) and C-3 (δ_C_ 158.4), one ketone C-1 (δ_C_ 173.8), one carboxylic acid carbonyl C-15 (δ_C_ 183.0), one methoxy carbon C-18 (δ_C_ 50.5) and one acetal C-4 (δ_C_ 111.9). HMBC correlations from H-16 to C-1, C-2, C-3 and from H-17 to C-2, C-3 and C-4 were observed (Figure 2). These correlations along with the additional HMBC correlation between H-18 and C-4 disclosed a substitued α,β-unsaturated 2,3-dimethyl-γ-lactone moiety, similar to that of the known butenolide 13-(2-hydroxy-3,4-dimethyl-5-oxo-2,5-dihydrofuran-2-yl) tridecanoic acid methyl ester and sinularones [28,29]. In the COSY spectrum, H-5, H-12, and H-13 showed correlations, respectively, to H-6, H-13, and H-14. In combination with the HMBC correlations between H-5 and C-4, C-6, C-7 in one hand, as well as H-14 and C-12, C-13, C-15 in the other hand, a carboxylic acid side chain from C-5 to C-15 was confirmed. In regard to the absolute configuration of carbon C-4, some of the known butenolides with α,β-unsaturated 2,3-dimethyl-γ-lactone moiety were reported to have *R* configuration at their stereogenic center and to show negative optical rotation, such as caulerpalide B [30]. Therefore, as compound **1** exhibited a negative optical rotation $\left({\left[a\right]}_{589}^{25}-40\right)$, we tentatively assumed C-4 to be 4*R* configuration. Consequently, compound **1** was identified as sinularone J, a new butenolide, analogue to sinularones [29].

Compound **2** was isolated as a colorless oil. High-resolution mass spectrometry (HRMS) analysis revealed a sodium adduct ion at *m*/*z* 433.2921 [M + Na]^+^ consistent with molecular formula C_24_H_42_O_5_ and incorporating four unsaturations. It provided a negative specific rotation ${\left[a\right]}_{589}^{25}-75$ (c 0.021, CH_3_OH). The NMR data of compound **2** (Table 2) were similar to those of **1** with the exception of the carboxylic acid side chain length. This information was supported by the molecular weight of **2** (*m*/*z* 433.2921) [M + Na]^+^, which is 84 amu more than that of **1** (*m*/*z* 349.1985) [M + Na]^+^ consistent with a difference of six methylenes. HMBC interactions from H-5 to C-4, C-6, C-7 and from H-20 to C-18, C-19, C-21 helped to establish the attachment of the carboxylic acid side chain made of 16 methylenes to the α,β-unsaturated 2,3-dimethyl-γ-lactone moiety (Figure 3). In regard to the absolute configuration of carbon C-4, we supposed C-4 to be 4*R* configuration as well. Therefore, compound **2** was identified as sinularone K, a structural analog of **1**.

Compound **3** was isolated as an amorphous powder. Its molecular formula was established as C_31_H_63_NO_9_P^+^ based on a protonated molecule ion at *m*/*z* 624.4235 [M + H]^+^ consistent with molecular formula C_31_H_62_NO_9_P, incorporating three unsaturations. HRMS/MS readily allowed the recognition of a phosphatidylcholine skeleton. MS/MS fragmentations decompose compound **3** to produce ions at *m*/*z* 510.4 [M + H]^+^ consistent with molecular formula C_26_H_56_NO_6_P (1-*O*-octadecyl-*sn*-glycero-3-phosphocholine), *m*/*z* 184.1 [M + H]^+^ consistent with molecular formula C_5_H_14_NO_4_P (protonated phosphocholine ion), *m*/*z* 125.0 [M + H]^+^ consistent with molecular formula C_2_H_5_O_4_P (protonated phosphate), *m*/*z* 86.1 [M + H]^+^ consistent with molecular formula C_5_H_11_N (phosphocholine head) that are diagnostic fragments for glycerophosphatidylcholine species [31]. Analysis of ^1^H NMR, HSQC and HMBC (Table 3) showed oxygenated methylene and methine proton signals C-1 (δ_H_ 3.59, 3.61; δ_C_ 70.2), C-2 (δ_H_ 5.17; δ_C_ 73.2), C-3 (δ_H_ 3.97, 4.03; δ_C_ 65.5) belonging to a glycerol unit, methylene proton signals attached to heteroatoms C-1’’ (δ_H_ 4.27; δ_C_ 60.5), C-2’’ (δ_H_ 3.65; δ_C_ 67.4) as well as a singlet signal integrated to nine methyl protons corresponding to three *N*-methyl groups (δ_H_ 3.23; δ_C_ 54.7) indicated the presence of choline. The presence of a linear alkyl chain was also shown by methylene proton signals (δ_H_ 1.25-1.54; δ_C_ 23.8-30.8). One methylene is attached to a heteroatom C-1’ (δ_H_ 3.43, 3.48; δ_C_ 72.7) and another one is a terminal methyl group C-18 (δ_H_ 0.90; δ_C_ 14.5). The connectivity between the glycerol unit at C-1 and the alkyl chain portion through the ether bond was determined from HMBC correlations between H-1 and C-2, C-3, C-1’ and between H-1’ and C-1 (Figure 4). Characteristic coupling between phosphorus and relevant protons (H-3 and H-1’’) were observed with ^1^H-^31^P HMBC correlations confirming the connectivity between the choline and the glycerol (C-3) units through phosphate. HMBC correlations from H-2 to C-1’’’ and from H-2’’’ to C-1’’’ indicated the presence of an ester-linked alkyl chain at C-2. The molecular formula of the residue at C-2 (C_5_H_6_O_3_) was deduced from HRMS/MS. The ^1^H NMR spectrum, along with HSQC and HMBC experiments (Table 3), showed one ester carbonyl C-1’’’ (δ_C_ 174.4), one carboxylic acid carbonyl C-5’’’ (δ_C_ 178.7) and three methylene carbons. Moreover, the COSY cross-peaks between H-2’’’, H-3’’’ and H-4’’’, in association with HMBC interactions of these protons to the carbonyl carbon C-5’’’ allowed to secure the carboxylic acid function. Therefore, the structure of **3** was determined as 1-*O*-octadecyl-2-pentanoyl-*sn*-glycero-3-phosphocholine.

Compound **4** was isolated as an amorphous powder. Its molecular formula was established as C_23_H_49_NO_8_P^+^ based on a protonated molecule ion at *m*/*z* 498.3188 [M + H]^+^ (calcd for C_23_H_49_NO_8_P^+^ 498.3190), incorporating 2 unsaturations. MS/MS fragmentations of compound **4** produce several diagnostic fragments ions of glycerophosphocholine [31], amongst which ions at *m*/*z* 258.1 [M + H]^+^ consistent with molecular formula C_8_H_20_NO_6_P (protonated glycerophosphocholine moiety)), *m*/*z* 184.1 [M + H]^+^ consistent with molecular formula C_5_H_14_NO_4_P (protonated phosphocholine ion), *m*/*z* 125.0 [M + H]^+^ consistent with molecular formula C_2_H_5_O_4_P (protonated phosphate), *m*/*z* 104.1 [M + H]^+^ and *m*/*z* 86.1 [M + H]^+^ consistent with molecular formula C_5_H_13_NO^+^ and C_5_H_11_N (phosphocholine head group fragments). Further information on the structure of the phospholipid was deduced from ^1^H NMR, COSY and HSQC data (Table 4). Analysis of the ^1^H NMR spectrum of compound **4**, along with HSQC experiment, showed the resonances of methylene and methine protons belonging to a glycerol moiety acylated at C-1 (δ_H_ 4.19, 4.13; δ_C_ 66.0), non-acylated at C-2 (δ_H_ 3.98; δ_C_ 69.4) and phosphorylated at C-3 (δ_H_ 3.90; δ_C_ 67.4). The shift in the oxygenated methylene protons signals observed at δ_H_ 3.59 and 3.61 (H-1) in **3** to δ_H_ 4.19 and 4.13 indicated the presence of an ester linkage instead of an ether linkage. The phosphocholine moiety was revealed by methylene proton signal attached to heteroatoms C-1’’ (δ_H_ 4.29; δ_C_ 60.1), C-2’’ (δ_H_ 3.65; δ_C_ 67.0) and a singlet signal integrated to nine methyl protons corresponding to three *N*-methyl groups (δ_H_ 3.23; δ_C_ 54.3). The presence of a linear alkyl chain was also observed in the ^1^H NMR spectrum along with characteristic proton and carbon signals arising from oxymethyl group (δ_H_ 3.34 and δ_C_ 56.9). Due to the low quantities obtained for **4**, the HMBC experiment could not be conducted. However, COSY correlations from H-2’ to H-3’ and HSQC correlation between H-3’ (δ_H_ 3.66) and C-3’ (δ_C_ 78.8) showed that the oxymethyl group is located at C-3’ (Figure 5). Therefore, the structure of **4** was determined as 1-*O*-(3-methoxy-tetradecanoyl)-*sn*-glycero-3-phosphocholine.

### 2.2. Molecular Networking

In an effort to fully capture the chemical space of this *Scopalina hapalia* specimen, molecular networking was used to complement the traditional chemical investigation. The former approach provides a method to rapidly assess the diversity and the chemical relatedness of metabolites based on similarities in MS/MS parent ion fragmentation patterns [32]. Fractions were prepared after removal of the major nonpolar lipids from *Scopalina hapalia* crude extract and analyzed by UHPLC-HRMS/MS in positive mode. Obtained MS/MS data were then used to generate molecular networks using the online plateform Global Natural Products Social Molecular Networking (GNPS). Molecular networks of the fractions (Figure 6A) revealed the presence of clusters related to different chemical classes (see Appendix A): polar lipids (lysophospholipids and fatty acids derivatives); amino acids derivatives (peptides and diketopiperazines); alkaloids (purines and β-carbolines) and terpenoids; reflecting at some points the results from the isolation procedure. The dereplication process highlighted three specific ion clusters (MN1–MN3). Interestingly, the isolated compounds **1** and **2** were detected in cluster MN1 (Figure 6B). Therefore, they were used as “seed molecules” in the analysis. Although no library hit was found for this cluster through the GNPS workflow, the dictionary of natural products database provided fruitful information for manual dereplication based on the molecular formulae proposed from HRMS and MS/MS fragmentation patterns. In that respect, two other 2,3-dimethyl-4-methoxybutenolides and two *β*-hydroxybutenolides (tetronic acid derivatives) were detected in cluster MN1. This partial annotation suggests that cluster MN1 aggregate different types of butenolide along with potential new butenolide analogues. The small number as well as the low amount of butenolide derivatives yielded by traditional detection and isolation techniques point to the probability that these compounds could be produced by the microbial symbionts of *Scopalina hapalia*. Typically associated with Gram-positive bacteria, butenolides are signaling molecules known to regulate morphological development and secondary metabolism, especially antibiotic production in the genus *Streptomyces* [33]. Commonly produced in low amounts (their biological concentration range from micro- to nano-molar), their isolation has proven difficult [34]. Furthermore, the analysis of the MS spectra from nodes belonging to clusters MN2 and MN3 revealed isotopic patterns that pointed toward brominated compounds (Figure 6C). Manual dereplication of cluster MN2 and MN3 based on the molecular formula matched, respectively, with six bromotyrosine alkaloids and two bromotyramine derivatives. Bromotyrosine-derived alkaloids are bioactive metabolites (antiviral, antibacterial, anticancer) typically found in sponges belonging to the Verongida order [35]. Given that this finding is based on the analysis of one specimen, additional work on *Scopalina* species would help to unambiguously demonstrate that these compounds can be detected in sponges belonging to the genus *Scopalina*. However, bromotyrosine derivatives have also been found in phylogenetically unrelated organisms like Tetractinellida sponges [36], ascidians [37,38] and the marine obligate bacterium *Pseudovibrio denitrificans* isolated from the marine sponge *Arenosclera brasiliensis* (order Haplosclerida) [39]. The occurrence of bromotyrosine across taxonomically distant group of organisms suggests that these metabolites might be produced by symbiont microorganisms. Hence, to prove the microbial origin of these metabolites (butenolides and bromotyrosines), future work based on molecular networking dereplication of the culturable microbial symbiont extracts need to be performed.

To sum up, in our preliminary investigation of marine sponge crude extracts for their anti-aging activities, the organic extract from *S. hapalia* inhibited the activity of the elastase and the protein kinase FynB. However, the screening of the fractions obtained subsequently did not show any biological activity. This lack of follow-up activity suggest that the strong inhibition observed for the crude extract might be the result of synergistic effect from metabolites. It is relevant to note that in spite of the multiple biological activities (antiproliferative, cytotoxic, antioxidant, antibacterial and antifungal) exhibited by either *S. ruetzleri* or *S. hapalia* crude extracts or fractions [13,14,15,16], there is only six metabolites reported from *S. ruetzleri*. Therefore, the fractionation of *S. hapalia* was pursued, independently of bioactivity, in order to gain a better understanding of its chemical composition. This work led to the isolation of fourteen metabolites, including two new butenolides derivatives sinularones J and K (**1** and **2**), one new phospholipid 1-*O*-octadecyl-2-pentanoyl-*sn*-glycero-3-phosphocholine (**3**) and one new lysophospholipid 1-*O*-(3-methoxy-tetradecanoyl)-*sn*-glycero-3-phosphocholine (**4**). The absolute configuration at C-4 of the new butenolides remains to be confirmed as we supposed it to be of *R* form. The compounds reported here, with the exception of **8**, represent new additions to the literature of this poorly studied genus. However, given the too low amounts obtained during isolation, most of the compounds (**1**–**4**, **6** and **7**, **10**–**14**) could not be evaluated against the selected molecular targets (elastase, tyrosinase, CDK7, FynB and proteasome). In particular, the new compounds were present in concentrations too low to be efficiently purified. To address these shortcomings, a collection of a new specimen of *Scopalina hapalina* is being considered. Compounds **5**, **8** and **9** were primary screened against CDK7, FynB and proteasome at three different concentrations (33 µg/mL, 3.3 µg/mL, 0.33 µg/mL) to identify inhibitors of these molecular targets. Compound **5** showed a weak inhibitory activity to proteasome, however only at the lowest concentration tested suggesting an eventual artifact. Consequently, compounds **5**, **8** and **9** were not further investigated for IC50 experimentation. These compounds were also evaluated for their activities in an elastase and tyrosinase bioassays at the concentrations of 1 and 10 µg/mL. None of these compounds displayed significant inhibition of elastase and tyrosinase activities. Compound **5** exhibited the highest inhibition by 7% at 1 µg/mL and 11% at 10 µg/mL for elastase bioassay; this activity is much lower compared to the active extract (inhibition by 44% at 1 µg/mL and 48% at 10 µg/mL in a cell-based elastase bioassay). Since these compounds were not significantly active, cell-based bioassays were not performed. As the current study was unable to identify bioactive compounds, an important question remains to be resolved for future studies, namely which metabolites are responsible for *S. hapalia* anti-elastase and anti-fynB activities, or are there synergistic effects of several compounds leading to the observed bioactivities. However, according to the literature, compounds **5**, **8**, **9**, **12**, **13** and **14** are already known to possess activities such as cytotoxicity, antifouling, antibacterial, immunomodulatory and quorum sensing sensors [23,27,40,41,42,43,44,45,46]. These bioactive compounds with antimicrobial and cytotoxic activities might explain the biological activities displayed by the *Scopalina* specimens. In addition, applying molecular networking for dereplication has proven to be useful. Based on the molecular network, several diketopiperazines and butenolides analogues were dereplicated. The most striking result to emerge from our data is the presence of bromotyrosine compounds in the fractions of *S. hapalia*. Additional work on the isolation of these predicted metabolites would be of great interest in order to confirm their presence. In conclusion, our results are encouraging and should be validated by further research on other *Scopalina* species.

## 3. Materials and Methods

### 3.1. General Experimental Procedure

NMR spectra were recorded on a Bruker Avance III-600 spectrometer (600 MHz for ^1^H and 150 MHz for ^13^C) equipped with de TCI Cryoprobe in 2.0 mm o.d. capillary tube at 300 K. Chemical shifts were referenced using the corresponding solvent signals (δ_H_ 3.31 and δ_C_ 49.00 for CD_3_OD 99.96%-d). The spectra were processed using 1D and 2D NMR TopSpin software.

High Resolution MS spectra of isolated compounds were measured using a Waters SYNAPT G2 HDMS TOF mass spectrometer with an electrospray ionization (ESI) source. Samples were solubilized in 300 µL of CH_3_OH and then diluted to 1/100 in a solution of 3 mM ammonium acetate in CH_3_OH. 

Optical rotations were measured on an MCP 200 Anton Paar modular circular polarimeter at 25 °C (CH_3_OH, c in g/100 mL) in a 10 × 5 mm i.d., 0.2 mL sample cell at 589 nm (Na D-line). 

MPLC separations were carried out on Buchi Sepacore flash systems C-605/C-615/C-660 and glass column (230 × 15 mm) packed with Macherey-Nagel MN Kieselgel silica gel (60−200 μm) or Acros Organic C-18 reverse phase. Analytical and semi-preparative HPLC were performed using either a Thermo Scientific Dionex Ultimate 3000 system equipped with a diode array detector (DAD) and a Corona detector or an Agilent 1200 system with DAD for analytical HPLC and a Gilson PLC 2020 with DAD for semipreparative HPLC. Analytical work was conducted using a Phenomenex Gemini C_18_ reversed-phased column (150 × 4.6 mm, 3 µm) and semi-preparative HPLC was undertaken using a Phenomenex Gemini C_18_ reversed-phased column (250 × 10 mm, 5 µm). UHPLC was carried out on Dionex Ultimate 3000 system connected to Bruker Impact II TOF mass spectrometer equipped with a Zorbax Eclipse Plus C_18_ (100 × 2.1 mm, 1.8 µm, Agilent) column. All solvents were analytical or HPLC grade and were used without further purification. The sponge was lyophilized with a CRYOTEC Cosmos 20K freeze dryer.

### 3.2. Animal Material

*Scopalina hapalia* (order Scopalinida, family Scopalinidae) was collected in May 2013 by scuba diving at the depths of 2–10 m around the south east coasts of Mayotte (Kani tip, GPS (12°57.624′ S; 45°04.697′ E). The sponge sample was levered off with a thin-bladed knife to prevent damage, transferred to a plastic bag and kept at −20 °C before being transported to the laboratory. The taxonomic identification was performed by Nicole de Voogd and a voucher specimen was preserved in 80% ethanol and is deposited at Naturalis Biodiversity Center, Leiden the Netherlands as RMNH POR.8376. *Scopalina hapalia* was originally described with some reservation as *Hymeniacidon hapalia* by Hooper et al., (1997) from Australia, however was transferred to the genus *Scopalina* by Alvarez and Hooper in 2010. The species is orange in coloration with a prominent spiky surface and its consistency is very fragile, soft and slimy. The skeleton is composed of ill-defined spongin fibres cored by spicules and obscured by granular material. The spicules are slightly curved styles with rough and uneven endings with a dimension of 338–*462*–566 × 5–*8*–10 µm (as min–*mean*–max). We compared our species with the related species *Scopalina australiensis* and *Scopalina rubra*. The present species is very similar to *Scopalina australiensis* from the Great Barrier Reef, however in addition to styles the spicules have oxeote modifications and are thinner and slightly larger in dimension. Another similar species is *Scopalina rubra* (originally described as *Ulosa rubra*) from Madagascar by Vacelet and Vasseur (1971), which is very close to our collection site. A type slide was examined (MNHN DJV32) and although the spicule dimensions are very similar to *S. hapalia*, namely 330–550 × 10–15 µm, the skeleton is very different. The skeleton is composed of regular thick multispicular spongin tracks and is not obscured by granular material. The outer morphology appears also very different, not as soft as our specimen with clear large oscules after preservation.

### 3.3. Extraction and Isolation Procedure

The lyophilized sponge (74.3 g, dry weight) was extracted overnight, once with Water to remove excess salt and three times using CH_3_OH/CH_2_Cl_2_ (1:1, v:v). The organic solvent was filtered and evaporated under reduced pressure to yield an orange/brown residue (7.82 g). A mass of 6.57 g of the extract was then partitioned between cyclohexane and aqueous CH_3_OH (10% water) (1:1, v:v). The methanolic layers were evaporated affording 2.76 g of a crude residue. A mass of 2.24 g of this extract was subjected to fractionation by medium pressure liquid chromatography MPLC. The sample was absorbed on silica gel C_18_ (<five times the mass of the sample), dry-loaded onto a sintered glass silica C_18_ a column (230 × 15 mm), and eluted using a gradient solvent system of H_2_O and CH_3_OH (10 mL∙min^−1^). Eighteen fractions of 45 mL each were collected. Fraction 7 (18.4 mg) was separated using reverse-phase C_18_ semipreparative HPLC, eluting with a gradient elution from 10% to 78% H_2_O/CH_3_CN (+0.1% formic acid F.A) over 55 min (3 mL∙min^−1^), yielding **1** (0.4 mg), **13** (0.1 mg) and **14** (0.3 mg). Fraction 16 (13.6 mg) was subjected to reverse-phase C_18_ semipreparative HPLC using a gradient elution from 78% to 97% H_2_O/CH_3_CN (+0.1% F.A) (3 mL∙min^−1^) yielding **12** (0.2 mg). Another portion of the crude extract CH_3_OH/CH_2_Cl_2_ (1:1, v:v) was fractionated by MPLC on silica gel using a combination of solvent with increasing polarity (2-methylpentane, EtOAc, CH_2_Cl_2_, CH_3_OH) yielding eleven fractions (15 mL∙min^−1^) (2-methylpentane/EtOAc 95:5; 2-methylpentane/EtOAc 90:10; 2-methylpentane/EtOAc 85:15; 2-methylpentane/EtOAc 80:20; 2-methylpentane/EtOAc 70:30; 2-methylpentane /EtOAc 50:50; EtOAc 100%; CH_2_CL_2_ 100%, 3 × CH_3_OH 100%). Fractions 4 and 5 eluted at 20 and 30 % EtOAc were combined to give a major fraction F4–F5 (~73.1 mg). An aliquot of this fraction (59.1 mg) was further separated using open column chromatography over silica gel, using isohexane with increasing proportions of EtOAc as eluent. Fractions eluted at 50% EtOAc were combined (13.7 mg) and subjected to reverse-phase C_18_ analytical HPLC. Elution was conducted with a linear gradient from 65% to 100% H_2_O/CH_3_CN (+0.1% F.A) over 30 min and held at 100% CH_3_CN (+0.1% F.A) for 20 min (0.7 mL∙min^−1^) yielding **2** (0.1 mg) and **11** (0.1 mg). An aliquot of F10-F11 (68.7 mg), eluted at 100% CH_3_OH, was separated using reverse-phase C_18_ semipreparative HPLC with a linear gradient from 5% to 100% H_2_O/CH_3_CN (+0.1% F.A) over 38 min and maintained at 100% CH_3_CN (+0.1% F.A) for 10 min (4.5 mL∙min^−1^) yielding **3** (0.1 mg), **4** (0.1 mg), **5** (0.1 mg), **6** (0.1 mg). An aliquot of fraction 6 (28.7 mg), eluted at 50% EtOAc, was separated *via* reverse-phase C_18_ semipreparative HPLC, eluting with an isocratic solvent system of H_2_O:CH_3_CN (+ 0.1% F.A) 10:90 (v:v) held for 45 min followed by 100% CH_3_CN + 0.1% F.A maintained for 10 min (4.5 mL∙min^−1^), yielding **7** (0.8 mg), **8** (4.0 mg), **9** (1.5 mg) and **10** (0.3 mg).

### 3.4. Compound Characterization

Sinularone J (**1**): yellow oil; ${\left[a\right]}_{589}^{25}-40$ (c 0.024, CH_3_OH); ^1^H and ^13^C NMR, see Table 1; HRMS *m*/*z* 349.1985 [M + Na]^+^ (calcd for C_18_H_30_O_5_Na^+^ 349.1985).

Sinularone K (**2**): colorless oil; ${\left[a\right]}_{589}^{25}-75$ (c 0.021, CH_3_OH); ^1^H and ^13^C NMR, see Table 2; HRMS *m*/*z* 433.2924 [M + Na]^+^ (calcd for C_24_H_42_O_5_Na^+^ 433.2924).

1-*O*-octadecyl-2-pentanoyl-*sn*-glycero-3-phosphocholine (**3**): amorphous powder; ^1^H and ^13^C NMR, see Table 3; ESI-MS *m*/*z* 624.4 [M + H]^+^ (6), 510.4 (1), 184.1 (100), 125.0 (3), 86.1 (4); HRMS *m*/*z* 624.4235 [M + H]^+^ (calcd for C_31_H_63_NO_9_P^+^ 624.4235).

1-*O*-(3-methoxy-tetradecanoyl)-*sn*-glycero-3-phosphocholine (**4**): amorphous powder; ^1^H and ^13^C NMR, see Table 4; ESI-MS *m*/*z* 498.3 [M + H]^+^ (40), 258.1 (4), 184.1 (100), 125.0 (4), 104.1 (13), 86.1 (5); HRMS *m*/*z* 498.3188 [M + H]^+^ (calcd for C_23_H_49_NO_8_P^+^ 498.3190).

### 3.5. Elastase and Tyrosinase Activity Assays

For the in vitro screening of the elastase enzyme activity, we used elastase from porcine pancreas (PPE) type IV and *N*-succinyl-Ala-Ala-Ala-*p*-nitroanilide as substrate; the assay was conducted as described before in Said Hassane et al. (2020) [47]. Elastatinal was used as positive control.

To evaluate the inhibitory potency of the extract and compounds against tyrosinase, the oxidation of l-DOPA to dopachrome was determined against mushroom tyrosinase (92 U/mL), as previously described [47]. Kojic acid was used as positive control. 

In both assays, experiments were performed in duplicates and negative control contained PBS and the substrate. Blanks contained all the assay components except the enzyme.

### 3.6. CDK7 Inhibition Assay 

CDK7 activity was evaluated using CDK7 (Crelux construct CZY3, PC11452). The inhibitory potency of the compounds against CDK7 was determined by using the ADP-Glo Kinase Assay (Promega, Madison, WI, USA) and CDKtide as substrate. The assay was carried out according to the method previously described [5]. An IC_50_ of 203 nM was determined for staurosporine using XLfit. It can generally be stated that the assay worked properly, as the control compound staurosporine, which was run in parallel to the assay, provided the expected results.

### 3.7. Fyn Kinase Inhibition Assay

FynB activity was evaluated using FynB wt (Crelux construct CTX4, PC09815-1). The inhibitory potency of the compounds against FynB was determined by using the ADP-Glo Kinase Assay (Promega) and Fyn kinase substrate (Enzo Life Sciences, P215). The assay was carried out according the method previously described [5]. An IC_50_ of 0.191 µM was determined for staurosporine using XLfit. In general, it can be stated that the assay worked properly, as the control compound staurosporine, which was run in parallel to the assay, exhibited good results.

### 3.8. Proteasome Inhibition Assay

Proteasome activity was evaluated using yeast proteasome (TUM Groll group) in its storage buffer: 20 mM Tris/HCl pH 7.5. The inhibitory potency of the isolated compounds to yeast proteasome was assessed by using the Fluorescence Intensity Assay and Suc-Leu-Leu-Val-Tyr-AMC as substrate (Enzo Life Sciences, BML-P802-0005). The assay was carried out according to the method previously described [5]. An IC_50_ of 0.0366 nM was calculated for ONX-0914 using XLfit. It can generally be stated that the Fluorescence Intensity Assay worked properly, as the control compound ONX-0914, which was run in parallel to the assay, showed the expected results.

### 3.9. Global Natural Product Social Molecular Networking

A molecular network was created using the online workflow (https://ccms-ucsd.github.io/GNPSDocumentation/, accessed on 23 June 2020) on the GNPS website (http://gnps.ucsd.edu, accessed on 23 June 2020) [32]. The data was filtered by removing all MS/MS fragment ions within ±17 Da of the precursor *m*/*z*. MS/MS spectra were window filtered by choosing only the top six fragment ions in the ±50 Da window throughout the spectrum. The precursor ion mass tolerance was set to 0.02 Da and an MS/MS fragment ion tolerance of 0.02 Da. A network was then created where edges were filtered to have a cosine score above 0.7 and more than six matched peaks. Furthermore, edges between two nodes were kept in the network if and only if each of the nodes appeared in each other’s respective top 10 most similar nodes. Finally, the maximum size of a molecular family was set to 100, and the lowest scoring edges were removed from molecular families until the molecular family size was below this threshold. The spectra in the network were then searched against GNPS’ spectral libraries. The library spectra were filtered in the same manner as the input data. All matches kept between network spectra and library spectra were required to have a score above 0.7 and at least six matched peaks. Mass spectrometry data of all H_2_O/CH_3_OH *Scopalina hapalia* fractions used in this study were added to the Public Massive datasets in GNPS (Massive ID MSV000085622).

## Figures and Tables

**Figure 1 marinedrugs-20-00186-f001:**
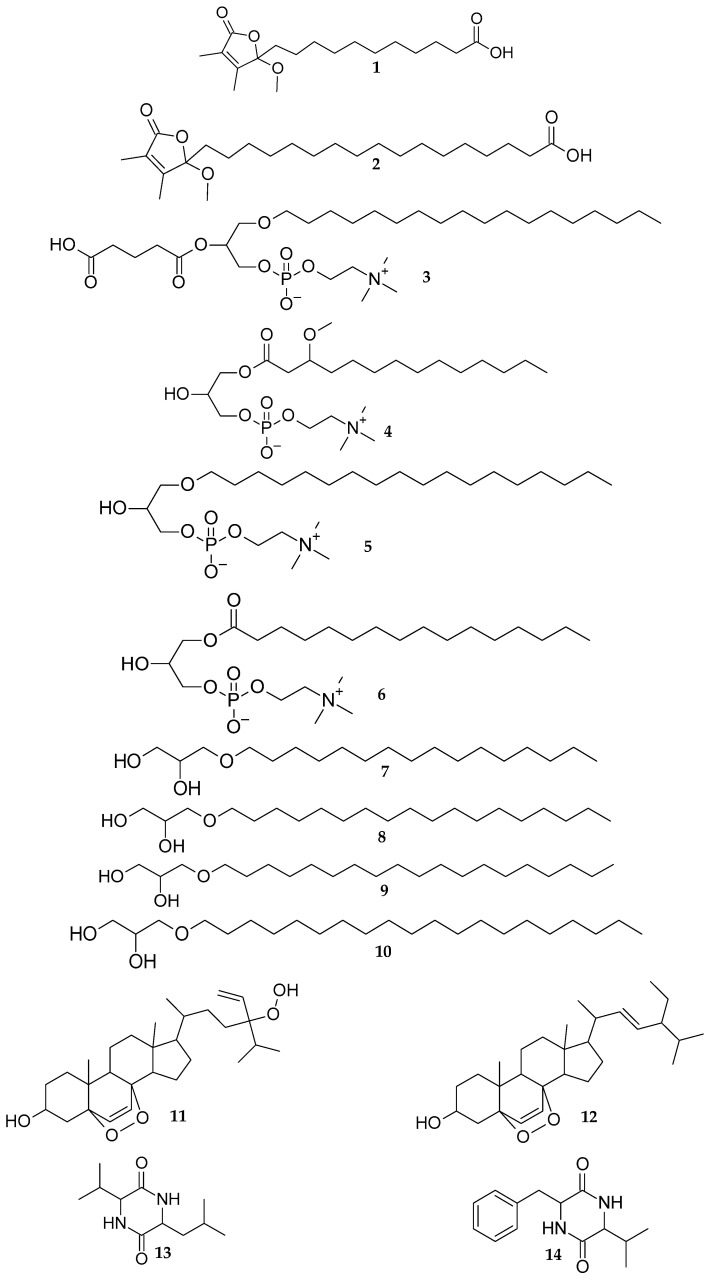
Chemical structures of new (**1**–**4**) and known (**5**–**14**) metabolites isolated from *Scopalina hapalia*.

**Figure 2 marinedrugs-20-00186-f002:**
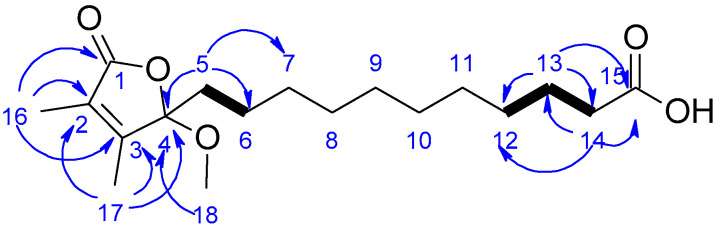
Key COSY (bold bonds) and HMBC (blue arrows) correlations for compound **1**.

**Figure 3 marinedrugs-20-00186-f003:**
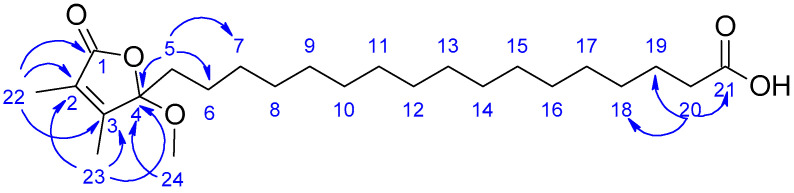
HMBC (blue arrows) correlations for compound **2**.

**Figure 4 marinedrugs-20-00186-f004:**
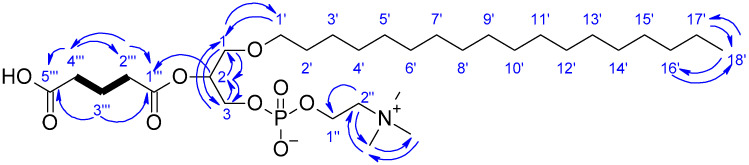
Key COSY (bold bonds) and HMBC (blue arrows) correlations for compound 3.

**Figure 5 marinedrugs-20-00186-f005:**
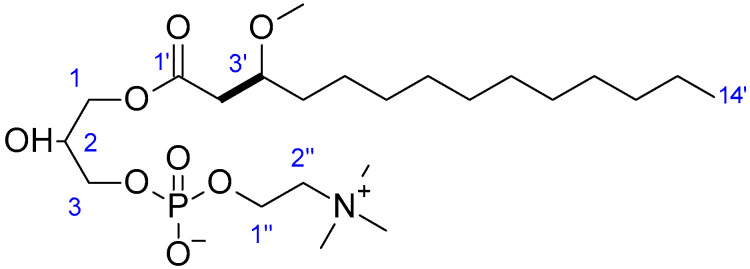
Key COSY (bold bonds) correlation for compound **4**.

**Figure 6 marinedrugs-20-00186-f006:**
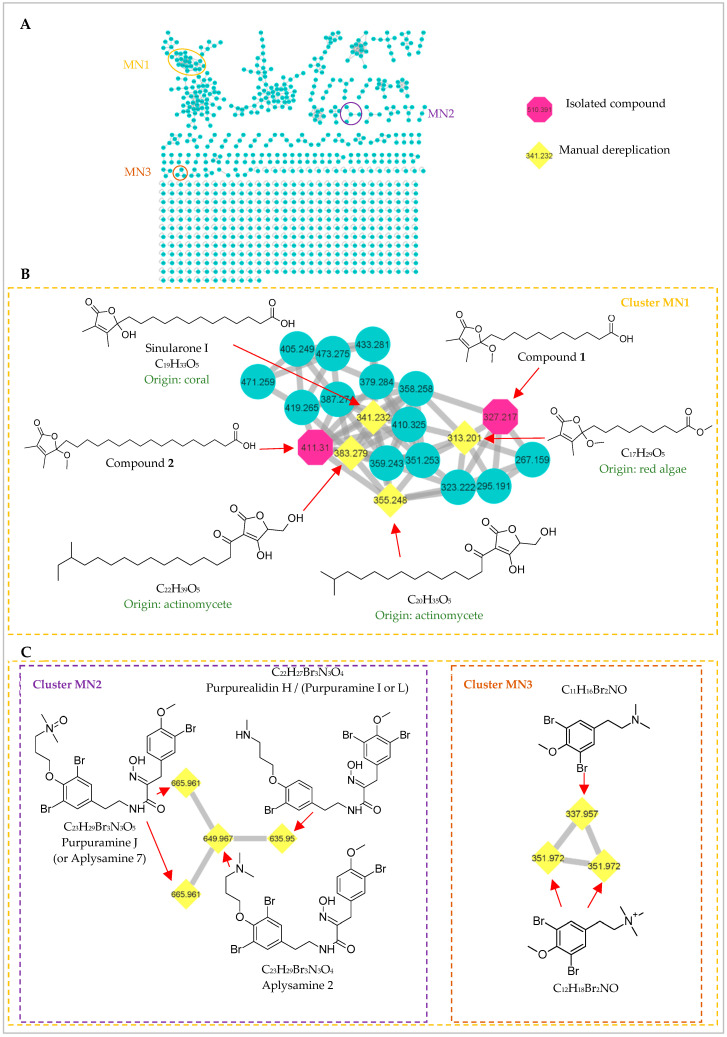
(**A**) Molecular Network constructed using MS/MS data from fractions of *Scopalina hapalia*, obtained after the removal of non-polar lipids of organic crude extract, with a cosine similarity cutoff of 0.7. Edge thickness corresponds to relative cosine score similarity between nodes. Annotated clusters are enlarged. (**B**) Cluster MN1 based on compounds **1** and **2** (octagonal nodes) related to butenolide derivatives. (**C**) Cluster MN2 and MN3 related to bromotyrosine metabolites.

**Table 1 marinedrugs-20-00186-t001:** NMR Spectroscopic data for **1** in CD_3_OD at 600 (^1^H) and 150 (^13^C) MHz.

No.	1		
	δ_C_, Type	δ_H_ (*J* in Hz)	HMBC, H → C
1	173.8, OC=O		
2	128.4, =CH		
3	158.4, =CH		
4	111.9, C		
5	36.5, CH_2_	1.95 (m) 1.73 (m)	C-4, C-6, C-7
6	23.8, CH_2_	1.24 (m) 1.13 (m)	
7–12	30.8–30.5, CH_2_	1.33–1.27 (m)	
13	27.7, CH_2_	1.59 (brqt 7.4)	C-12, C-14, C-15
14	39.1, CH_2_	2.15 (brt7.8)	C-12, C-13, C-15
15	183.0, C		
16	8.2, CH_3_	1.84 (q, 1.2)	C-1, C-2, C-3
17	10.9, CH_3_	1.90 (q, 1.2)	C-2, C-3, C-4
18	50.5, CH_3_	3.07 (s)	C-4

**Table 2 marinedrugs-20-00186-t002:** NMR Spectroscopic data for **2** in CD_3_OD at 600 (^1^H) and 150 (^13^C) MHz.

No.	2		
	δ_C_ ^a^, Type	δ_H_ (*J* in Hz)	HMBC, H → C
1	173.8, OC=O		
2	128.4, =CH		
3	158.4, =CH		
4	111.9, C		
5	36.6, CH_2_	1.96 (m) 1.73 (m)	C-4, C-6, C-7
6	23.9, CH_2_	1.24 (m) 1.14 (m)	
7–18	31.0–30.6, CH_2_	1.33–1.27 (m)	
19	27.8, CH_2_	1.59 (qt, 7.5)	
20	39.2, CH_2_	2.16 (t, 7.5)	C-18, C-19, C-21
21	182.9, C		
22	8.3, CH_3_	1.83 (q, 1.1)	C-1, C-2, C-3
23	10.9, CH_3_	1.90 (q, 1.1)	C-2, C-3, C-4
24	50.5, CH_3_	3.07 (s)	C-4

^a^ δ_C_ were determined from HSQC and HMBC experiments.

**Table 3 marinedrugs-20-00186-t003:** NMR Spectroscopic data for **3** in CD_3_OD at 600 (^1^H) and 150 (^13^C) MHz.

No.	3		
	δ_C_ ^a^, Type	δ_H_ (*J* in Hz)	HMBC, H → C
1	70.2, CH_2_	3.59 (1H, dd 10.9, 5.9) 3.61 (1H, dd 10.9, 4.7)	C-2, C-3, C-1’
2	73.2, CH	5.17 (1H, m)	C-1, C-3, C-1’’’
3	65.5, CH_2_	3.97 (1H, dt 11.1, 6.1 ^b^) 4.03 (1H, ddd 11.1, 5.7, 4.2 ^b^)	C-1, C-2
1’	72.7, CH_2_	3.43 (1H, dt 9.3, 6.6) 3.48 (1H, dt 9.3, 6.7)	C-1, C-2’, C-3’
2’	30.8, CH_2_	1.54 (2H, brqt 6.8)	
3’	27.6, CH_2_	1.35–1.25 (30H, ov ^c^)	
4’–15’	30.7, CH_2_	1.33–1.27 (m)	
16’	33.2, CH_2_	1.59 (qt, 7.5)	
17’	23.8, CH_2_	2.16 (t, 7.5)	
18’	14.5, CH_3_	0.90 (3H, t 7.1)	C-16’, C-17’
1’’	60.5 (d 5.4) ^b^, CH_2_	4.27 (2H, m ^b^)	
2’’	67.4 (m) ^b^, CH_2_	3.65 (2H, m)	C-1’’, N^+^(CH_3_)_3_
N^+^(CH_3_)_3_	54.7 (m) ^b^, CH_3_	3.23 (brs)	C-2’’, N^+^(CH_3_)_3_
1’’’	174.4, C		
2’’’	34.6, CH_2_	2.42 (2H, brt 7.5)	C-1’’’, C-4’’’
3’’’	22.1, CH_2_	1.90 (2H, brqt 7.5)	C-1’’’, C-5’’’
4’’’	35.9, CH_2_	2.31 (2H, brt 7.3)	C-2’’’, C-5’’’
5’’’	178.7, C		

^a^ δ_C_ were determined from HSQC and HMBC experiments; ^b 1^H−^31^P couplings; ^c^ overlapping signals.

**Table 4 marinedrugs-20-00186-t004:** NMR Spectroscopic data for **4** in CD_3_OD at 600 (^1^H) and 150 (^13^C) MHz.

No. ^a^	4		
	δ_C_ ^b^, Type	δ_H_ (*J* in Hz)	COSY, H → H
1	66.0, CH_2_	4.19 (dd, 10.9; 4.7) 4.13 (dd, 10.9; 5.9)	
2	69.4, CH	3.98 (m)	
3	67.4, CH_2_	3.90 (m)	
1’	no	-	
2’	39.9, CH_2_	2.54 (dd, 15.3; 7.3) 2.51 (dd, 15.3; 5.6)	3’
3’	78.8, CH	3.66 (m)	2’
4’	34.5, CH_2_	1.53 (m)	
5’–12’	33.2, CH_2_	1.35–1.25 (ov ^c^)	14’
13’	23.4, CH_2_	2.16 (t, 7.5)	
14’	14.1, CH_2_	0.90 (t, 7.1)	13’
OCH_3_	56.9, CH_3_	3.34 (s)	
1’’	60.1, CH_2_	4.29 (m)	2’’
2’’	67.0, CH_2_	3.65 (m)	1’’
N^+^(CH_3_)_3_	54.3, CH_3_	3.23 (s)	

^a^ All assignments are based on ^1^H NMR and 2D NMR measurements (COSY, HSQC); ^b^ δ_C_ were determined from HSQC experiment; ^c^ overlapping signals.

## Data Availability

The data presented in this study are available in the main text and the supplementary material of this article. 1D and 2D NMR raw data of compounds 1–4 are made freely available at https://doi.org/10.5281/zenodo.6301017.

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
