# Peer review of "New Metabolites from the Marine Sponge Scopalina hapalia Collected in Mayotte Lagoon"

_marinedrugs, 2022, doi:10.3390/md20030186_

Round 1

Reviewer 1 Report

good improvements from the last version. congrats.

Author Response

Thank you very much for your comments.

Reviewer 2 Report

The paper titled "New metabolites from the marine sponge Scopalina hapalia collected in Mayotte lagoon" by Charifat Said Hassane et al. describes the isolation, structural determination and GNPS analysis of four new compounds (two butenolides, one phospholipid and one lysophospholipid). I think the topic of this paper is worthy of publication in Marine Drugs. On the other hand, I think that the structural determination for the butenolides still have majour problem as indicated below.

I can not understand the description for the specific rotations and the stereochemistry of compounds 1 and 2.
If "another lipid compound" is present in the sample, as described in the paper, then both compounds 1 and 2 are mixtures, and the discussion about their structure and GNPS analysis itself is not valid. (This could also jeopardize the credibility of the paper).
Some of the related butenolides, which has a similar skeleton to the butenolides isolated in this study, have R-stereochemistry at C-4 position. (e.g., (+)-caulerpalide B and (-)-caulerpalide B were isolated from the green alga Caulerpa racemosa.*) So, the stereochemistry at C-4 of compounds 1 and 2 are not necessarily S, but can also be R.
The authors should measure the CD spectrum because it will give the absolute stereochemistry clearly, and I hope the authors will submit again with conclusive data.

  • D.-C. Li et al., Three new butenolides from the green alga Caulerpa racemosa var. turbinata. Chem. Biodivers., 2020 May; 17(5) e2000022.

There are a few minor errors that need to be corrected. Some of them are shown below.

# There is an error in naming for comppound 4. Please correct "1-O-(3-methox-tetra---" to "1-O-(3-methoxy-tetra---". 

# In p. 2, line 57. Please correct "aldo" to "aldol".

# in p. 5, Table 2. Please correct "182.9, CH2" to "182.9, C".

Author Response

Thank you very much for your comments.

Please find enclosed an answer to your questions.

Reviewer 3 Report

In the present manuscript, Gauvin-Bialecki and co-workers decribe the isolation and structural elucidation of four novel natural products,  two butenolides (sinularone J and K), one lysophospholipid, and one phospholipid from the marine sponge Scopalina hapalia. The work has been executed accurately and the structures of the new natural products have been satisfactorily characterized based on their corresponding spectroscopic data including 2D NMR spectroscopy. The manuscript has been prepared very carefully and the obtained compounds are interesting due to their pharmacological potential. In conclusion, this very good piece of research is strongly recommended for publication in Marine Drugs.

Author Response

Thank you very much for your comments.

Round 2

Reviewer 2 Report

I understand the authors' argument. The corrections have been well made. I believe this paper will be of interest to readers of the Marine Drugs and would like to recommend its acceptance with the following minor corrections.

In lines 27, 427, 498 and 499. Correct "3-methox" to "3-methoxy".

In lines 484 to 520. Correct "1H" and "13C" to "1H" and "13C".

In line 432. Correct "N-succinyl" to "N-succinyl", "-p-nitroanilide" to "-p-nitroanilide".

There may be some minor errors that I have overlooked, so please recheck carefully.

This manuscript is a resubmission of an earlier submission. The following is a list of the peer review reports and author responses from that submission.